# Text messaging reminders for influenza vaccine in primary care: protocol for a cluster randomised controlled trial (TXT4FLUJAB)

Emily Herrett,[1] Tjeerd van Staa,[1,2] Caroline Free,[1] Liam Smeeth[1]

▶ Prepublication history and additional material is available. To view please visit the journal (http://dx.doi.org/10.1136/bmjopen-2013-004633).

[1]London School of Hygiene and Tropical Medicine, London, UK
[2]Division of Pharmacoepidemiology and Clinical Pharmacology, Department of Pharmaceutical Sciences, Faculty of Sciences, Utrecht Institute for Pharmaceutical Sciences, Utrecht University, Utrecht, The Netherlands

**Correspondence to**
Dr Emily Herrett;
emily.herrett@lshtm.ac.uk

## ABSTRACT

**Introduction:** The UK government recommends that at least 75% of people aged under 64 with certain conditions receive an annual influenza vaccination. Primary care practices often fall short of this target and strategies to increase vaccine uptake are required. Text messaging reminders are already used in 30% of practices to remind patients about vaccination, but there has been no trial addressing their effectiveness in increasing influenza vaccine uptake in the UK.

The aims of the study are (1) to develop the methodology for conducting cluster randomised trials of text messaging interventions utilising routine electronic health records and (2) to assess the effectiveness of using a text messaging influenza vaccine reminder in achieving an increase in influenza vaccine uptake in patients aged 18–64 with chronic conditions, compared with standard care.

**Methods and analysis:** This cluster randomised trial will recruit general practices across three settings in English primary care (Clinical Practice Research Datalink, ResearchOne and London iPLATO text messaging software users) and randomise them to either standard care or a text messaging campaign to eligible patients. Flu vaccine uptake will be ascertained using routinely collected, anonymised electronic patient records. This protocol outlines the proposed study design and analysis methods.

**Ethics and dissemination:** This study will determine the effectiveness of text messaging vaccine reminders in primary care in increasing influenza vaccine uptake, and will strengthen the methodology for using electronic health records in cluster randomised trials of text messaging interventions. This trial was approved by the Surrey Borders Ethics Committee (13/LO/0872). The trial results will be disseminated at national conferences and published in a peer-reviewed medical journal. The results will also be distributed to the Primary Care Research Network and to all participating general practices.

**Trial registration number:** This study is registered at controlled-trials.com ISRCTN48840025, July 2013.

## Strengths and limitations of this study

- This trial is the first to evaluate the effectiveness of text messaging for influenza vaccine reminders in English primary care.
- Text messaging is cheap, quick and has shown to be effective for appointment reminders.
- The study requires minimal input from practices as it uses routinely collected electronic health records to gather vaccine uptake data.
- There may be some contamination if general practices in the standard care arm choose to send a text message.

Health Service and to the UK as a whole. Influenza vaccine is safe and effective but is required annually because the circulating strain of the virus changes each year. In the UK in 2012, the Chief Medical Officer recommended that at least 75% of elderly people (aged 65+) and 75% people under 65 with certain chronic conditions (eg, chronic heart disease, diabetes, asthma, etc) should be vaccinated.[1]

While many primary care practices are achieving these targets for elderly patients (74% vaccinated in 2011/2012), those set for patients with chronic conditions are not being met and have shown no substantial increases in the past decade[2]; vaccine uptake in 2011/2012 was 51.6% across the risk groups (see online supplementary table S1). Barriers to influenza vaccination in the UK include the lack of recommendation for vaccination by a healthcare worker,[3] lack of awareness of eligibility for vaccination,[4] low perceived susceptibility to and severity of influenza,[3–5] beliefs that the effectiveness of the vaccine is low, and beliefs about its side effects, safety and pain.[3 5] Strategies to increase flu vaccine uptake are required.

Several randomised trials have demonstrated the effectiveness of flu vaccine reminders delivered to patients by letter, postcard

## INTRODUCTION

Influenza morbidity and mortality causes a substantial financial burden to the National

or telephone.[6] [7] However, the use of text messaging in the NHS for appointment reminders is increasing as it is cheap, quick and effective. Text messaging is already used in roughly 30% of practices to remind patients about their flu vaccine,[8] but there has been no trial addressing its effectiveness in this context. Trials of text messaging in the USA have shown some success as vaccine reminders,[9] but whether there is an effect in UK primary care is unknown.

Therefore, we are performing a randomised trial of a text messaging flu vaccine reminder in patients aged under 65 who have a chronic condition (as described in online supplementary table S1). We will randomise whole primary care practices to either a text messaging campaign to eligible patients or to standard care (chosen as the comparator group because practices currently employ a variety of methods to encourage at-risk patients for influenza vaccine).[8]

## AIMS AND OBJECTIVES
The aims of the study are (1) to develop the methodology for conducting cluster randomised trials of text messaging interventions using routine electronic health records and (2) to implement a cluster randomised trial to test the effectiveness of using a text messaging influenza vaccine reminder in achieving an increase in influenza vaccine uptake in patients aged 18–64 with chronic conditions compared with standard care.

Specific objectives are: (1) to evaluate the effect of text messaging influenza vaccination reminders in patients under 65 with chronic conditions; (2) to evaluate the feasibility of recruiting and randomising practices to a text messaging intervention; (3) to evaluate the feasibility of practice delivery of a text message intervention to eligible patients within a primary care setting and (4) to evaluate the feasibility of ascertaining practice data regarding text message delivery and flu vaccine uptake.

## METHODS
### Study design
A cluster randomised trial[10] [11] of a text messaging influenza vaccination reminder in primary care.

### Study population
This study is taking place in English primary care, with general practices recruited from three settings: (1) the Clinical Practice Research Datalink (CPRD),[12] a primary care database based on Vision software and covering 8% of the UK population; (2) ResearchOne, a primary care database based on TPP SystmOne software[13] and covering 7.6% of the UK population and (3) iPLATO text messaging software users in London.[14]

### Inclusion criteria
Eligible practices must currently use a text messaging system to communicate with patients about issues other than influenza vaccination (CPRD and London practices must use iPLATO,[14] ResearchOne practices must use TPP SystmOne text messaging software. Practices using other text messaging software will be excluded). Practices that used text messages for influenza vaccination reminders in the 2012/2013 influenza season will be excluded.

### Recruitment
Eligible practices in the CPRD (n=40) will be identified using the electronic health record and invited to the trial. All TPP SystmOne practices (n=2073) and London-based iPLATO practices (n=460) will be invited to the trial by the Primary Care Research Network. London iPLATO practices will also be contacted by iPLATO and TPP SystmOne practices will be targeted through SystmOne communications. The Primary Care Research Network[15] will be involved in practice recruitment to ensure that the target sample size is reached.

### Sample size calculation
The sample size calculation is based on a comparison between influenza vaccine uptake in the intervention group, in which a text message influenza vaccination reminder will be sent to eligible patients, and influenza vaccine uptake in the standard care group. A cluster-level analysis of the practice-specific proportions will be undertaken and the sample size calculation estimates the number of practices required for the study. A systematic review of reminders for influenza vaccination reported that reminders (of any type) for influenza vaccine among adults increased the uptake, with an OR of 1.66, 95% CI 1.31 to 2.09).[16] However, a text messaging vaccine reminder trial among children in the USA found only a 9% increase (RR=1.09, 95% CI 1.04 to 1.15).[9] Therefore, we have chosen to power our study for a risk ratio of 1.075, representing an increase from 54% (uptake in the CPRD 2012/2013) to 58%.

At 90% power and 5% significance, with an ICC by general practice of 0.024 and an average of 750 eligible patients per practice, we will require 100 practices (50 per arm) to identify a 7.5% increase in vaccine uptake from 54% to 58%. To account for differences in the number of eligible patients per practice and the proportion with mobile phone numbers recorded, we have chosen to recruit and randomise 150 practices to the study.

### Assignment of interventions
Practices will be randomised on a 1:1 basis to the intervention and standard care groups. We will use block randomisation within each setting. In the CPRD and ResearchOne, we will stratify by region. The allocation sequence will be generated by an independent statistician who will be blinded to practice name and the allocation sequence will be concealed from the study team. Once practices have been allocated, the trial coordinator will inform practices about their allocation and distribute study materials.

## Blinding

Practices in this study will be randomised to intervention or standard care groups, but general practice staff will not be blinded. However, academic investigators and trial statisticians will manage and review data without knowledge of the allocation.

## Interventions

Practices will be allocated to an intervention or standard care arm. Those in the intervention arm will be asked to send an influenza vaccination text message reminder to their under 65 at-risk patients. This includes patients who are under 65 with chronic heart disease, chronic neurological disease, diabetes, chronic kidney disease, chronic liver disease, chronic respiratory disease and immunosuppression, as set out by the Chief Medical Officer.[17] Most practices will identify eligible patients based on their electronic medical records using their established systems. Practices use a standard set of Read codes (available from Primis[18]) to determine patient eligibility for influenza vaccination.

The message content that practices will be asked to send is shown in box 1. This single text message was designed to be a pragmatic, uncomplicated intervention that would not discourage practices from participating, but that aims to address some of the key barriers to vaccination.[3] It was specifically designed to target more than one factor influencing behaviour: first, the text message represents a recommendation from the general practitioner (GP); second, it indicates the severity of flu and that the patient is susceptible to flu; third, it acts as a prompt to patients who might have simply forgotten to make an appointment.

Practices in the intervention group will receive guidance notes regarding delivery of the message (content, timing, eligible patients) and an incentive payment of £200. Practices will be instructed to continue with any other aspect of their seasonal flu campaign as planned, in addition to the text message. An online questionnaire will be sent to practices in the intervention arm to ascertain feedback from practice staff about delivery of the intervention.

Practices allocated to the standard care group will be asked to continue with their seasonal flu campaign as planned.

### Substudy for patient feedback

A substudy will be conducted in two participating intervention practices. Each practice will send an anonymous questionnaire to patients who were targeted with the text message reminder. This questionnaire will address patient acceptability of the text message.

## Outcomes

The aims of the study are (1) to develop the methodology for conducting cluster randomised trials of text messaging interventions using routine electronic health records and (2) to implement a cluster randomised trial to test the effectiveness of using a text messaging influenza vaccine reminder in achieving an increase in influenza vaccine uptake in patients aged 18–64 with chronic conditions compared with standard care.

To address aim 2, we will examine the proportion of eligible patients who received the influenza vaccination by 31 December 2013 (primary outcome). This will be stratified by risk group, age and sex. A secondary outcome will be the proportion of eligible patients who were sent a text message to remind them about flu vaccine.

To address aim 1, we will measure the following as secondary outcomes:
► Practice recruitment rate (based on the number of practices invited across three trial settings, and the number recruited);
► The proportion of intervention practices and standard care practices that sent a text message to eligible patients;
► The proportion of practices reporting any problems with delivering the message;
► The availability of data to examine text message receipt and vaccine uptake in each risk group;
► The time and cost required to gather data.

In the patient substudy, we will also examine the acceptability of text messaging to patients, by measuring the proportion of patients who reported problems with the text message.

## DATA COLLECTION METHODS

All patient-level outcomes will be evaluated through analysis of routinely collected data. In practices that contribute data to the CPRD or ResearchOne, relevant data will be extracted from the database using specified Read codes. Within London practices using iPLATO software, data will be extracted by the clinical software supplier of the practice or the practices themselves.

We will extract data on age, sex, clinical risk group (based on historical data from the medical record and prescription data), vaccination reminder type (text, letter, phone call, face to face), vaccination uptake, death, transfer out of practice, flu and flu-like illness, and hospital admissions. Uptake and patient outcomes will be analysed from 1 September to 31 December 2013. All of these data items are routinely collected by general practices and stored in their electronic health record, and follow-up for practices will be complete using this method. Time and cost of data collection using these methods will be estimated. All data will be

---

**Box 1 Recommended text message content in the trial intervention arm.**

Hello *PATIENT NAME*, to reduce your risk of serious health problems from flu, we recommend vaccination. Call *PRACTICE PHONE NUMBER* to book. *PRACTICE NAME*.

---

stripped of personal identifiers before being supplied to the study team, and will be stored securely as defined in the protocol.

We will record the number of practices approached, recruited and analysed. An online questionnaire, emailed to each practice in the intervention group, will ascertain practice acceptability of the intervention. The postal questionnaire to patients in the substudy will estimate patient acceptability.

## STATISTICAL METHODS

A cluster-level analysis will be performed using practice-specific proportions as observations. We will compare vaccine uptake in the intervention and standard care groups using a t test, with the size of clusters taken into account. A series of descriptive statistics will describe the methodological outcomes.

### Sensitivity and subgroup analyses

Our primary analysis will be an intention to treat analysis. However, to account for any contamination between the standard care and intervention arms, we will carry out a per-protocol analysis.

Where available, we will also measure the difference in vaccine uptake, comparing practices that used the exact wording of the message in the study protocol with practices that used an alternative message. We hypothesise that practices using our message (based on behavioural theory) will have a higher uptake than an alternative designed by the practice. As this will be a non-randomised comparison, we will adjust for confounders.

Finally, we will compare the effectiveness of the text message based on the time of day that it was sent to patients. We hypothesise that messages sent in the late afternoon will have more effect than those sent at earlier times of the day, when patients may not have the time to respond.

### Hawthorne effect

The study design allows an evaluation of the generalisability of the study population; practices that take part in the study can be compared with other practices outside of the trial that contribute data to ResearchOne and CPRD. We will test whether participation in the trial changes the behaviour of practices in their use of text messaging (Hawthorne effect).

This study will provide evidence regarding the effectiveness of text messaging reminders for influenza vaccine in patients under 65 with chronic conditions. The methodology here will be applied to future cluster randomised trials of text messaging interventions within electronic heath records.

## ETHICS AND DISSEMINATION
### Ethical issues
#### Approvals

The study protocol (V.2.7, 11/02/2014) has received a favourable opinion from the National Research Ethics Service Committee—Surrey Borders (REC number 13/LO/0872) and has also received assurance from participating Primary Care Trusts/Clinical Commissioning Groups. All minor and major amendments to the protocol will be approved by the REC, and participating general practices will be notified of any protocol amendments relevant to them.

#### Informed consent

This is a cluster trial and informed consent will be obtained by the trial coordinator from GPs, who will act as the guardian of the cluster. A signed and dated informed consent form is required for participation in the trial. The risks and benefits of participation will be explained and GPs are free to decline to participate in the trial. Individual consent will not be sought. This is justified because participating practices already use text messaging to communicate with patients, although not systematically for influenza. Patients within these practices will have the opportunity to opt out of receiving messages if they do not wish to be contacted in this way.

#### Patient confidentiality

All patient-level data accessed in this study will be stripped of personal identifiers.

#### Access to data

The data will be accessed only by authorised persons from the London School of Hygiene and Tropical Medicine and research governance authorities to ensure that the study is being carried out to acceptable standards. All will have a duty of confidentiality and no data will be disclosed outside the research site. All patient-level and practice-level information will be confidential.

#### Burden and risk to practices

Text messaging is widely used in everyday primary care and the only substantive change from non-study is the random allocation of a defined text message regarding influenza vaccine rather than the practice's own seasonal flu campaign. Patients can opt out of receiving text messages from their practices, and as part of this study, we will monitor patient and practice complaints regarding the text message. Patients registered at practices in the intervention arm will receive one text message regarding their flu vaccine to minimise irritation to patients.

Participating practices must spend time identifying eligible patients and sending a text message. We are reducing this burden by (1) asking practices to use established procedures and software to identify eligible patients for vaccination. This is a task that they would perform annually in non-study conditions; (2), we are including only practices that are familiar with the text

messaging software, which minimises time spent in familiarisation with the software and sending messages; (3) additional support is available to any practice through a guidance document and/or dedicated support helpline and (4) we are providing an incentive payment (£200) to practices in the intervention group to encourage participation.

Practices may experience increased demand for influenza vaccination appointments or run out of vaccine supplies sooner than expected. The increase in uptake is unlikely to be large and will help the practice to achieve their Quality and Outcomes Framework target for vaccination, for which they receive payment.

## Benefits for practices

Practices in the intervention arm will receive a £200 incentive to participate. If the text message is effective, then increased uptake will result in additional incentive payment to GPs by the government and will help them reach their Quality and Outcomes Framework targets. The results of this trial will allow practices to make an informed decision about spending of their budget on the seasonal influenza campaign.

## Dissemination plans

The results of this trial will be published according to the guidelines of the CONSORT statement. The results will be published in a peer reviewed medical journal and presented at national conferences. The results will also be distributed to the Primary Care Research Network and to all participating general practices. Authorship for all publications will be based on the criteria defined by the International Committee of Medical Journal Editors.[19]

**Acknowledgements** The authors thank the LSHTM Clinical Trials Unit, Tim Collier, Noclor, Public Health England, the Cabinet Office, the Primary Care Research Network, the Department of Health, the Clinical Practice Research Datalink, ResearchOne and iPLATO for their guidance and support.

**Funding** This work was supported by the Wellcome Trust grant number 098504/Z/12/Z.

**Competing interests** None.

**Ethics approval** Surrey Borders NREC.

**Provenance and peer review** Not commissioned; externally peer reviewed.

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
