## [Reviewer comments · BMJ Open]

Some articles will have been accepted based in part or entirely on reviews undertaken for other BMJ Group journals. These will be reproduced where possible.

ARTICLE DETAILS

TITLE (PROVISIONAL)	Text messaging reminders for influenza vaccine in primary care: protocol for a cluster randomised controlled trial (TXT4FLUJAB)
AUTHORS	Herrett, Emily; van Staa, Tjeerd; Free, Caroline; Smeeth, Liam

VERSION 1 - REVIEW

REVIEWER	Elyse O. Kharbanda HealthPartners Institute for Education and Research
REVIEW RETURNED	18-Feb-2014

GENERAL COMMENTS	The authors should expand their references and discussion of current barriers to influenza vaccination in the UK and the methods that will be used to identify patients with chronic medical conditions In this manuscript, the authors describe their plans for implementing a cluster randomized study of text messages to increase influenza vaccination in the UK. The proposed study is ambitious in scope and potential public health impact. The authors should be commended for the efforts to detail and disseminate their study aims, analytic strategy and potential study limitations a priori. I have several comments regarding the proposed methodology. First, the authors provide little background regarding why adults with chronic medical conditions have low vaccination coverage in the UK. This would be helpful to justify whether text messages are likely to be effective. Text messages may help to increase vaccine uptake if the current barriers are lack of awareness or competing priorities (people are busy and forget to get vaccinated). However, if the primary barriers to adult vaccination in the UK are negative beliefs regarding vaccine efficacy and/or safety, a single text message is unlikely to have any effect. Second, the authors report that they are planning to have practices send a single text message to eligible adults at intervention clinics in order to "minimise irritation to patients". However, they have powered their study based on a similar study that my colleagues and I conducted in NYC. In our study, our modest increase in influenza vaccination in children was achieved by sending a median of 5 text messages per family. I doubt that a single message would have the same effect. I would suggest that the authors change this aspect of their approach. Third, the authors state they will identify adults with chronic medical conditions "based on their electronic medical record." How patients recommended for vaccination are identified is key to the success of this intervention and should be described in more detail. Will chronic conditions be identified from problem lists, medications or diagnoses at past visits? Please also provide additional data on the validity of
---

	the classification system that will be used. Finally, will data be available on whether a patient received influenza vaccine in the prior season? This variable is likely to be the strongest predictor of whether a patient receives influenza vaccine in the current season. Thus, it would be an important variable to include in the process of randomization, to ensure balance between intervention and control sites.
--	--

REVIEWER	Carolyn R. Ahlers-Schmidt University of Kansas School of Medicine - Wichita, United States
REVIEW RETURNED	19-Feb-2014

GENERAL COMMENTS	This protocol describes an important study to evaluate the use of text message reminders for patients with chronic conditions. However, the paper is missing many of the components recommended in the SPIRIT checklist and lacks sufficient detail in many of the areas. I recommend the authors revised the manuscript using the SPIRIT checklist. In addition, the paper needs proof reading, additional references (e.g. page 4, line 39), and further discussion of possible confounders. For example, what if practices use multiple methods for vaccination reminders, or send multiple text message reminders?
---

REVIEWER	Peter Szilagyi University of Rochester School of Medicine and Dentistry USA
REVIEW RETURNED	17-Mar-2014

GENERAL COMMENTS	This paper describes a proposed clustered RCT based in 3 groups of primary care practices to test the impact of sending a single text message reminder to eligible patients under 65 years on receipt of influenza vaccination. 150 practices from the 3 networks, all of which already utilize text messaging software, will be randomized to study (1 text message sent by the practice) or standard of care control group; outcomes will be influenza vaccination rates and process measures (and a small substudy from 2 practices will assess patient feedback). Standard statistical methods will assess impact. The study is important because influenza vaccination rates in this population and throughout much of the world are suboptimal. While many practices use text messaging (mostly to remind patients of upcoming scheduled appointments), they have not been tested in the UK for impact on influenza vaccination rates. The clustered RCT design is appropriate, the sample size is large, and outcomes are readily accessible. Some minor aspects of the study design are not optimal or not well described. First, it is unclear how many practices currently send text or other reminders for flu vaccination (the RCT design will account for it but if most practices send such reminders, the impact will be dampened). Second, the protocol calls for all practices in at least one of the large networks to be invited to participate, and it may not be possible to ascertain response rates using this method. Third, it is unclear exactly how practices will identify eligible patients based on specific diagnoses. Fourth, studies of patient reminders suggest that
--

	more than one reminder is more effective than a single reminder, yet this study proposes a single reminder in order to not irritate patients. The proposed intervention may be a weak one. Fifth, the study is powered overall well but effect sizes for subgroup comparisons are not shown. These are all correctable limitations. Overall the study is important, innovative and should contribute to the prevention field.
--	---

VERSION 1 – AUTHOR RESPONSE

Reviewer Name Elyse O. Kharbanda
 Institution and Country HealthPartners Institute for Education and Research
 Please state any competing interests or state 'None declared': None declared

The authors should expand their references and discussion of current barriers to influenza vaccination in the UK and the methods that will be used to identify patients with chronic medical conditions
 We have expanded the material on barriers and methods as suggested.

In this manuscript, the authors describe their plans for implementing a cluster randomized study of text messages to increase influenza vaccination in the UK. The proposed study is ambitious in scope and potential public health impact. The authors should be commended for the efforts to detail and disseminate their study aims, analytic strategy and potential study limitations a priori.
 We thank the reviewer for these comments.

I have several comments regarding the proposed methodology. First, the authors provide little background regarding why adults with chronic medical conditions have low vaccination coverage in the UK. This would be helpful to justify whether text messages are likely to be effective. Text messages may help to increase vaccine uptake if the current barriers are lack of awareness or competing priorities (people are busy and forget to get vaccinated). However, if the primary barriers to adult vaccination in the UK are negative beliefs regarding vaccine efficacy and/or safety, a single text message is unlikely to have any effect.
 We agree with the reviewer that there was limited background regarding the barriers to vaccination and have now added a short section on this to the Introduction. We have also added a sentence to the Methods to describe how the content of our text message might have addressed three of these barriers.

Second, the authors report that they are planning to have practices send a single text message to eligible adults at intervention clinics in order to "minimise irritation to patients". However, they have powered their study based on a similar study that my colleagues and I conducted in NYC. In our study, our modest increase in influenza vaccination in children was achieved by sending a median of 5 text messages per family. I doubt that a single message would have the same effect. I would suggest that the authors change this aspect of their approach.
 We agree with the reviewer that there is some evidence that messages targeting multiple rather than single factors influencing behaviour are more effective. In particular Free et al's previous systematic review has shown that whilst simple reminders delivered by text message have very small benefits for appointment attendance they have no benefits in increasing adherence to taking medication. For this reason our message was designed to address three factors, a GP recommendation for flu vaccine, the importance for health of flu vaccine and a reminder. We agree a more complex and intensive intervention might be more effective but for ease of implementation we chose a single message for this first trial which is intended to assess the feasibility of conducting a trial of a text message intervention within the electronic health records. We have added a sentence to the Methods to justify our simple intervention. Our plans for the future are to assess the effectiveness of a more complex

text message intervention in the same setting.

Third, the authors state they will identify adults with chronic medical conditions "based on their electronic medical record." How patients recommended for vaccination are identified is key to the success of this intervention and should be described in more detail. Will chronic conditions be identified from problem lists, medications or diagnoses at past visits? Please also provide additional data on the validity of the classification system that will be used.

The Chief Medical Officer in the UK recommends that certain groups of 'at-risk' patients are targeted by the seasonal influenza campaign. Primary care practices use a common set of Read codes to define patients who are at risk. These are supplied by Primis (part of the University of Nottingham) and are the codes against which practices' flu vaccine uptake are assessed. We have now added a sentence to the Methods section to describe this.

Finally, will data be available on whether a patient received influenza vaccine in the prior season? This variable is likely to be the strongest predictor of whether a patient receives influenza vaccine in the current season. Thus, it would be an important variable to include in the process of randomization, to ensure balance between intervention and control sites.

Data on previous vaccination will be available to us through the medical record. However, as our protocol randomises entire practices to either the intervention or standard care group, previous vaccination (a patient-level factor) cannot be used in the randomisation process.

Reviewer Name Carolyn R. Ahlers-Schmidt

Institution and Country University of Kansas School of Medicine - Wichita, United States

Please state any competing interests or state 'None declared': None declared

This protocol describes an important study to evaluate the use of text message reminders for patients with chronic conditions. However, the paper is missing many of the components recommended in the SPIRIT checklist and lacks sufficient detail in many of the areas. I recommend the authors revised the manuscript using the SPIRIT checklist.

We thank the reviewer for highlighting this omission. We have now added further content and a SPIRIT checklist in the supplementary material.

In addition, the paper needs proof reading, additional references (e.g. page 4, line 39), and further discussion of possible confounders. For example, what if practices use multiple methods for vaccination reminders, or send multiple text message reminders?

We have added further references to our manuscript and it has been carefully proof read.

Our protocol allocated practices to either intervention or a standard care group using randomisation, the purpose of which was to distribute practice characteristics evenly between our two groups. Therefore, for the main analysis, we hope that practices using a variety of different reminder methods are evenly balanced between the groups. Sub-group analyses, which will be non-randomised comparisons, will be adjusted for confounders.

Reviewer Name Peter Szilagyi

Institution and Country University of Rochester School of Medicine and Dentistry

USA

Please state any competing interests or state 'None declared': No conflicts

This paper describes a proposed clustered RCT based in 3 groups of primary care practices to test the impact of sending a single text message reminder to eligible patients under 65 years on receipt of

influenza vaccination. 150 practices from the 3 networks, all of which already utilize text messaging software, will be randomized to study (1 text message sent by the practice) or standard of care control group; outcomes will be influenza vaccination rates and process measures (and a small substudy from 2 practices will assess patient feedback). Standard statistical methods will assess impact.

The study is important because influenza vaccination rates in this population and throughout much of the world are suboptimal. While many practices use text messaging (mostly to remind patients of upcoming scheduled appointments), they have not been tested in the UK for impact on influenza vaccination rates. The clustered RCT design is appropriate, the sample size is large, and outcomes are readily accessible.

Some minor aspects of the study design are not optimal or not well described. First, it is unclear how many practices currently send text or other reminders for flu vaccination (the RCT design will account for it but if most practices send such reminders, the impact will be dampened).

The reviewer is correct that there is likely to be some contamination; we attempted to reduce this by excluding practices that sent text messages in the previous influenza season. We also increased our target sample size to account for some contamination. We are aware that any contamination will reduce our ability to identify an effect of text messaging and in our analysis we will measure and report exactly how many practices in the standard care group used text messaging. We have added this as a secondary outcome to clarify.

Second, the protocol calls for all practices in at least one of the large networks to be invited to participate, and it may not be possible to ascertain response rates using this method.

Our response rate will be based on the number of practices in the TPP network, and the number of practices that consented and participated in the trial. This has been clarified in the Outcomes section of the manuscript.

Third, it is unclear exactly how practices will identify eligible patients based on specific diagnoses. We apologise for the lack of clarity in this point and have now added a sentence to explain that practices use a standard set of Read codes to identify patients to target in their seasonal flu campaigns. These are made available by Primis, part of the University of Nottingham.

Fourth, studies of patient reminders suggest that more than one reminder is more effective than a single reminder, yet this study proposes a single reminder in order to not irritate patients. The proposed intervention may be a weak one.

As we discussed in our response to Reviewer 1, we agree that a more complex text messaging intervention may be effective than a single message. However, our study was also intended to assess the feasibility of conducting a trial of a text message intervention within electronic health records. We felt that our simple intervention encouraged practices to take part, allowing us to assess the effectiveness of a single text message reminder and also to determine the feasibility of this kind of randomised controlled trial. We have added a sentence to the Methods to justify our simple intervention.

Fifth, the study is powered overall well but effect sizes for subgroup comparisons are not shown. Our study was powered to detect the main effect of the text messaging intervention. It was not powered for these subgroup analyses, and therefore our study may miss a real effect of message content or timing, should any effects exist.

These are all correctable limitations.

Overall the study is important, innovative and should contribute to the prevention field.